# Time to recovery and predictors of severe community-acquired pneumonia among pediatric patients in Debre Markos referral hospital, North West Ethiopia: A retrospective follow-up study

**Belayneh Mengist**[1]*, **Mulugeta Tesfa**[1], **Bekalu Kassie**[2]

**1** Department of Public Health, College of Health Sciences, Debre Markos University, Debre Markos, Ethiopia, **2** Department of Midwifery, College of Health Sciences, Debre Markos University, Debre Markos, Ethiopia

* belaynehmengist2008@gmail.com

**Data Availability Statement:** The data sets used for this study are available as Supporting Information.

## Abstract

### Introduction

Globally, pneumonia is a major cause of morbidity and mortality among children which leads to over 156 million episodes and 14.9 million hospitalizations each year. Besides this fact, the recovery time and predictors of children's hospitalization related to severe community-acquired pneumonia is not well known. Therefore, the aim of this study was to estimate the median time to recovery and its predictors among severe community-acquired pneumonia patients admitted to the pediatric ward, Debre Markos referral hospital, North West Ethiopia.

### Methods

An institution-based retrospective follow-up study was employed among 352 records of children who were admitted starting from January 2016 to December 2018. Patients' charts were retrieved using a structured data extraction tool. Cox proportional hazard model assumption and model fitness was checked. Stratified Cox regression was fitted as a final model. Hazard ratio with its 95% confidence interval was used and P-value < 0.05 was considered as a statistically significant association.

### Result

The overall median recovery time was 4 days IQR (3–7). Recovery rate from severe community acquired pneumonia was 16.25 (95% CI: 14.54–18.15) per 100 person day observation. Age (AHR; 0.94 95% CI (0.90–0.98)), being stunted (AHR; 0.62 95% CI (0.43–0.91)), presence of danger sign at admission (AHR; 0.61 95% CI (0.40–0.94)), late presentation to seek care(AHR; 0.64 95% CI (0.47–0.88)) and co-morbidity (AHR; 0.45 95% CI (0.35–0.58)) were significant predictors of recovery time.

**Funding:** The author(s) received no specific funding for this work.

**Competing interests:** The authors have declared that no competing interests exist.

**Abbreviations:** AHR, Adjusted Hazard Ratioh; AOR, Adjusted Odds Ratio; ARTI, Acute Respiratory Tract Infection; CI, Confidence Interval; DMRH, Debre Markos Referral Hospital; EDHS, Ethiopia Demographic and Health Survey; HIV, Human Immune Virus; IMNCI, Integrated Management of Neonatal and Childhood Illness; ICCM, Integrated Community Case Management; IQR, Inter Quartile Range; PCV, Pneumococcal Conjugate Vaccine; SCAP, Severe Community-Acquired Pneumonia; SDG, Sustainable Development Goals.

## Conclusion

The median recovery time from severe community-acquired pneumonia was long so that measures to reduce recovery time should be strengthened.

## Introduction

Globally, pneumonia is a major cause of morbidity and mortality among children which leads to over 156 million episodes and 14.9 million hospitalizations each year [1]. Pneumonia is an acute respiratory tract infection (ARTI) that affects the parenchymal tissues of the lungs [2]. During normal breathing, small sacs in the lungs called alveoli fill with air. When children contract pneumonia the alveoli fill with pus and fluid, restricting breathing and making it painful [3]. It is a common infection of the lungs affecting millions of people worldwide. World health organization (WHO) has defined pneumonia solely on the basis of clinical findings obtained by visual inspection and on the timing of the respiratory rate [4].

Community-acquired pneumonia (CAP) is an infection that begins outside the hospital or is diagnosed within 48 hours after admission to the hospital in a person who has not resided in a long-term care facility for 14 days or more before admission [5]. Hospital-acquired pneumonia is pneumonia that occurs more than 48 hours after admission and without any antecedent signs of infection at the time of hospital admission [6]. CAP is a leading infectious disease requiring hospital admission and constitutes a major burden on health care resources [7].

Clinical factors of illness at admission affect the prognosis of pediatric pneumonia. A study conducted on risk factors for a poor outcome among children admitted with clinically severe pneumonia to a university hospital in Rabat, Morocco showed that impaired consciousness, cyanosis, and pallor were independent risk factors for an adverse outcome [8].

Another study noted that as a median time of clinical stability increases, the severity of pneumonia also increases significantly [9]. A prospective observational study on the Severity of Pneumonia in under-five children also showed that the length of hospitalization days varied for children with hypoxemia and without hypoxemia [10]. The type of drug prescribed for severe community-acquired pneumonia (SCAP)patients had a significant effect on the recovery time of the disease [11].

Pneumonia has been reduced significantly after the introduction of the pneumococcal conjugate vaccine (PCV) [12]. However, it is a major cause of morbidity and mortality of children globally in general and among low-income countries in particular. Although pneumonia is highly treatable in most patients, it can be a serious medical condition requiring hospitalization and become life-threatening particularly among children. Studies were conducted on the prevalence, associated factors, and determinants of pneumonia among under-five children [5]. However, those studies were conducted mainly among under-five children and did not determine the predictors of recovery time. Recovery time and its predictors of children's hospitalization related to SCAP are not well known. In addition, as far as our literature search, there was no study reporting estimated survival period of recovery. Determining up-to-date predictors of recovery time and period of hospitalization by including children whose age is under-15 is crucial. Therefore, this study was conducted to estimate the median time to recovery and its predictors from severe community-acquired pneumonia among patients admitted to the pediatric ward, Debre Markos referral hospital (DMRH), North West Ethiopia.

## Methods

### Study design

An institution-based retrospective follow-up study was employed.

## Study area and period

This study was conducted at Debre Markos referral hospital which is the only referral hospital in the East Gojjam zone and found in Debre Markos, the town of the administrative Zone. According to information obtained from the administrative office of the hospital, it serves more than 3.5 million populations in its catchment area. The hospital provides services for children less than 15 years in a separate ward of pediatrics following the national guideline which was adapted from WHO. The hospital admits an average of 3293 children per year, with 31 available beds. In this hospital, children with severe community-acquired pneumonia were admitted into the pediatric ward and further diagnosis and treatment were provided by pediatricians, general practitioners, and nurses. Patients were differentiated from other lower respiratory tract infections clinically. There is no pediatric intensive care unit (ICU) in the hospital; emergency pediatric conditions are treated in the ward.

The study was conducted from January 2016 to December 2018, among children who were admitted with SCAP, and registration charts were retrieved from March 15 to 30, 2019.

## Population

**Source population.**   All children admitted to the hospital with severe community-acquired pneumonia during the study period.

**Study population.**   All children admitted to the hospital with severe community-acquired pneumonia whose charts were available during data collection time.

## Eligibility criteria

**Inclusion criteria.**   The inclusion criteria were children who were from 1 month to15 years and admitted to the pediatric ward with severe community-acquired pneumonia that is pneumonia with any one of oxygen saturation <90% or central cyanosis or severe respiratory distress or inability to drink or breastfeed or vomiting everything, altered consciousness, and convulsions (WHO, 2013) during the study period.

**Exclusion criteria.**   The exclusion criterion was children who were admitted to the hospital with incomplete medical records.

## Sample size determination and procedure

**Sample size determination.**   To determine sample size various predictors significantly associated with the outcome variable were considered. Accordingly, the sample size was determined using a double population proportion formula by considering, P1: the proportion of exposed (62.3%), P2: the proportion of non exposed (47.4%), ratio(1), power (80%), 95% confidence level using vaccination status as a major predictor based on the study conducted in Kersa district, South West Ethiopia [5] using the Kelsey formula in Epi info version 7 statistical software. The sample size for this study was 352.

**Sampling procedure.**   The study participants were selected from the registration book. The medical records of children who were admitted with severe community-acquired pneumonia from January 2016 to December 2018 were selected. A total of 793 children were recorded from the registration book of which 721 completed cards were available. Then 352 cards were sampled using a simple random sampling technique by a computer-generating method. Finally, data were extracted from the selected medical charts.

## Variable of the study

**Dependent variable.**

✓  Time to recovery from severe community-acquired pneumonia

### Independent variables

- ✓ Age of children
- ✓ Sex of children
- ✓ Residence
- ✓ Presence of concomitant disease (co-morbidity)
- ✓ Nutritional status of children (stunting, wasting, underweight)
- ✓ Time elapsed to seek care (duration)
- ✓ History of previous ARTI
- ✓ Vaccination status of children
- ✓ Health insurance
- ✓ Clinical presentation during admission/danger sign (impaired consciousness, abnormal body movement, vomiting everything)
- ✓ Breastfeeding
- ✓ Drug regimen

### Operational definitions

**Recovery:** children improved from SCAP as declared by the clinician.

**Event:** recovery from an illness during the study period.

**Survival time:** defined as the time starting from the date of admission to recovery from SCAP determined for each participant.

**Censored:** children referred, died, or discharged for any reason without recovery during the study period.

**Pediatrics:** children from 1month to 15 years of age.

**Co-morbidity:** any disease condition (acute or chronic) present at admission in addition to SCAP which includes hyperactive airway disease (childhood asthma), retroviral infection, tuberculosis, acute gastroenteritis, pertussis, anemia, meningitis, measles, bronchitis, heart failure, and urinary tract infection.

**Danger sign:** loss of consciousness, abnormal body movement, vomiting everything, convulsions, and inability to feed in addition to SCAP [13].

**Vaccination status:** full vaccination status was defined as children who had completed all forms of vaccinations; partially vaccinated (children who had taken at least one dose of PCV) and non vaccinated (children who had never vaccinated PCV and other vaccines).

**Respiratory distress** in this study context was considered as the presence of grunting and head nodding.

### Data collection procedure

Patient records were retrieved using their unique registration number identified in the electronic hospital database. Charts and records were reviewed and data were extracted using a standardized English data extraction form.

### Data collection tool

A structured data extraction tool was developed by considering study variables such as socio-demographic, nutritional, and clinical predictors from patients' charts.

### Data quality assurance

Data were collected from patients' charts by two experienced BSc nurses who were working in the pediatric ward. A pretest was done in the hospital on 5% of charts and modification of the checklist was made. One day training was given for both data collectors and supervisors concerning the data collection tool and data collection process. Data quality was also assured by designing proper data abstraction tool and through continuous supervision. All collected data were checked for completeness and clarity.

### Data processing and analysis

Data were entered using Epi-Data version 3.02 and analysis was done using STATA Version14 statistical software. Data were cleaned and edited before analysis. The data were described in terms of central tendency (median) and dispersion (IQR), in frequency distribution, and graphically for categorical data.

Days were used as a time scale to calculate time to recovery. The outcome of each participant was dichotomized into censored or event (recovered). In this study, an acute form of malnutrition was assessed among the participants using the WHO Z-score charts of weight for height or length. Z-scores of between -3SD and $\leq$ -2SD were equated to moderate acute malnutrition and Z-scores of $\leq$ -3SD were equated to severe acute malnutrition, also the presence of kwashiorkor was considered as severe acute malnutrition.

The life table was used to estimate cumulative survival probabilities after admission. Kaplan Meier survival curve and log-rank test were used to describe the survival function. Bivariable Cox-proportional hazards regression model was fitted for each predictor. Those variables having a p-value $\leq$ 0.25 in the bivariable analysis were selected. Then, the further variable selection was undertaken using a stepwisebackward variable selection approach using a p-value $\leq$ 0.25 as a cut point. Cox proportional hazard assumption was checked for each covariate using Schoenfeld residuals tests, and graphically using a log-log plot of survival. Predictor drug regimen violates the proportional hazard assumption and the Cox-Snell residual suggested that the Cox proportional hazard model (Cox regression) does not fit the data adequately. Therefore an extension of the Cox proportional hazard model was required for this data. Since predictors were tested as not time-dependent, the stratified Cox regression model was adequate to be used.

To check the no-interaction assumption of the stratified Cox regression model, the likelihood ratio (*LR*) test was used by comparing log-likelihood statistics for the interaction and the non-interaction model. This *LR* test statistic had approximately achi-square distribution under the null hypothesis that the non-interaction model was correct. Therefore, stratified Cox regression with no interaction model based on drug regimen as the stratified variable and seven predictors which hold the proportional hazard assumption was carried out as the final model for this study. The hazard ratio with its 95% confidence interval was used to measure the strength of association and the p-value < 0.05 was used to identify the statistically significant association.

### Ethics statement

Ethical clearance was obtained from the ethical review committee of Debre Markos University, College of Health Sciences (Ref. No: HSC/983/16/18). A formal letter was submitted to Debre

Markos referral hospital and permission was assured. All information collected from patient cards was kept strictly confidential and names of patients were not included in the checklist. Consent was not requested as it was a retrospective study.

## Result

### Socio-demographic characteristics

A total of 793 severe community-acquired pneumonia patients were admitted to the pediatric ward of Debre Markos referral hospital during the study period. Of which 352 children were included in the study. The median age of the participants was 1.17 years IQR (0.67, 2.17) (Table 1).

### Baseline nutritional and vaccination status

The majority of children 308(87.5%) were on exclusive breastfeeding for the first six months. Anthropometric measurements were compared with WHO child growth standards to classify the nutritional status of children. From the overall study participants, 21(5.97%), 15(4.26), and 14(3.98%) of them were categorized under severe underweight, severe stunting, and severe wasting respectively. Of the study participants 65.6%, 27%, and 7.4% were fully vaccinated, partially vaccinated and non-vaccinated respectively.

### Clinical characteristics

The diagnosis was made mainly based on clinical findings; oxygen saturation was recorded and a chest x-ray was conducted for patients who had strong indications. Aetiologic micro-organisms of SCAP (bacteria or virus) were not identified. Two third(68.5%) of children were febrile at the admission of which nearly one quarter had a high-grade fever. Of all study participants, 9.09% of them developed danger signs at admission. The mean respiratory rate of children at admission was 61 with a standard deviation of 12.9.

Among children admitted due to SCAP, half (50.57%) of them had co-morbidity. The most frequent comorbidity was hyperactive airway disease(childhood asthma)followed by acute

**Table 1. Socio-demographic characteristics, the median time to recovery, and comparison of survival time among SCAP patients, DMRH, North West Ethiopia, 2016 to 2018.**

| Variable | Frequency (percent) (n = 352) | Median recovery time Point estimate(95%CI) | Log rank $x^2$ value(df) | P-value |
|---|---|---|---|---|
| **Age (years** | | | | |
| <1 | 171(48.58) | 4(4–5) | 16.93(2) | 0.0002 |
| 1–5 | 150(46.61) | 4(3–4) | | |
| 5–14 | 31(8.81) | 7(6–8) | | |
| **Sex** | | | | |
| Male | 185(52.56) | 5(4–5) | 7.37(1) | 0.0066 |
| Female | 167(47.44) | 4(3–4) | | |
| **Residence** | | | | |
| Rural | 176(50) | 5(4–5) | 24.73(1) | <0.0001 |
| Urban | 176(50) | 4(3–4) | | |
| **Insurance** | | | | |
| Insured | 98(27.84) | 6(4–7) | 20.28 (1) | <0.0001 |
| Not insured | 254(72.16) | 4 | | |

df- degree of freedom.

gastroenteritis and about 4.3% of children had more than one co-morbidities at admission (**Table 2**).

## Treatment regimen and outcome

Of all children who were treated as an inpatient, more than half (59.66%) received crystalline penicillin. The majority of children (88.9%) recovered from their illness,5.11% left against medical advice,3.69% referred to other institutions and 2.27% have died. The median time to death for those who died was 6 days. Patients were treated with crystalline penicillin, ceftriaxone, ampicillin, and gentamicin based on the treatment guideline which is recommended by WHO [14]. Non-invasive respiratory support was given for 46 (13%) of children who were in respiratory failure.

## The median time to recovery

The estimated median time to recovery from SCAP for all observations was 4 days IQR (3–7). The median recovery time of children from SCAP varied among various categories of socio-demographic predictors. For instance, the median recovery time of under one and one to five years of children was 4 days whereas above five years children were 7 days (**Table 1**).

The median recovery time of children from SCAP was different regarding different categories of nutritional status and clinical characteristics (**Table 3**).

## The incidence rate of recovery from SCAP

From 352 study participants, 313 (88.9%) of children developed an event and the rest 39 (11.1%) children were censored observations. The lowest and the highest length of follow-up were 1 and 34 days respectively, and the total person-time risk was 1923. The overall recovery rate from SCAP was16.25 per 100 person day(95% CI: 14.54–18.15) observation. The recovery

**Table 2. Clinical features and co-morbidities of SCAP patients admitted to DMRH, North West Ethiopia, 2016 to 2018.**

| Variable | Frequency (n = 352) | Percentage (%) |
|---|---|---|
| **Signs** | | |
| Oxygen saturation <90% | 203 | 57.7 |
| Severe respiratory distress | 46 | 13.1 |
| High grade fever(>38.5ºc) | 86 | 24.4 |
| Vomiting everything | 3 | 0.8 |
| Altered consciousness | 21 | 5.9 |
| Abnormal body movement | 8 | 2.3 |
| **Co-morbidities** | | |
| Hyperactive airway disease | 71 | 20.2 |
| Retroviral infection | 12 | 3.41 |
| Tuberculosis | 20 | 5.7 |
| Acute gastroenteritis | 36 | 10.2 |
| Pertussis | 13 | 3.7 |
| Meningitis | 12 | 3.4 |
| Anemia | 10 | 2.8 |
| Heart failure | 12 | 3.4 |

Some patients had multiple features at admission.

**Table 3. Median time to recovery and comparison of survival time among SCAP patients by nutritional and clinical characteristics, DMRH, North West Ethiopia, 2016 to 2018.**

| Variable | Frequency (percent) (n = 352) | Median recovery time | Log rank | p-value |
|---|---|---|---|---|
| | | Point estimate(95%CI) | $x^2$ value(df) | |
| **WFA** | | | | |
| Normal | 277(78.69) | 4 | 27.65(1) | <0.0001 |
| Under wt. | 75 (21.31) | 5 (4–8) | | |
| **HFA** | | | | |
| Normal | 297(84.38) | 4 | 28.75(1) | <0.0001 |
| Stunting | 55 (15.62) | 6(5–10) | | |
| **WFH** | | | | |
| Normal | 294 (83.52) | 4 | 17.46(1) | <0.0001 |
| Wasting | 58 (16.48) | 6(5–7) | | |
| **Duration** (days) | | | | |
| ≤5(early presenters) | 259 (73.58) | 4(3–4) | 53.99(1) | <0.0001 |
| >5(late presenters) | 93 (26.42) | 6(5–8) | | |
| **Past history of ARTI** | | | | |
| No | 229 (65.06) | 4 | 12.97(1) | 0.0003 |
| Yes | 123 (34.94) | 5(4–6) | | |
| **Danger sign** | | | | |
| No | 320 (90.91) | 4 | 15.05(1) | 0.0001 |
| Yes | 32 (9.09) | 8(5–12) | | |
| **Co-morbidity** | | | | |
| No | 174 (49.43) | 3(3–4) | 74.28(1) | <0.0001 |
| Yes | 178 (50.57) | 6(5–7) | | |
| **Drug regimen** | | | | |
| Crystalline penicillin | 210 (59.66) | 4(3–4) | 41.11(2) | <0.0001 |
| Ceftriaxone | 117 (33.24) | 4(4–5) | | |
| Ampicillin & gentamicin | 25 (7.10) | 11(7–18) | | |

WFH: Weight for age, HFA: Height for age, WFH: Weight for height, df: degree of freedom.

rate among male and female children was 14.64 per 100 person day (95% CI: 12.52–17.12) and 18.26 per 100 person day (95% CI: 15.61–21.37) respectively.

## Survival estimates for time to recovery

The survival status of children with SCAP was estimated by the Kaplan-Meier survival curve. The curve tends to decrease rapidly within the first ten days indicating that most children recovered from the disease within this time (**Fig 1**). The survival estimates of SCAP patients were varied in relation to comorbidity, danger signs, and duration to seek care (**Figs 2–4**).

## Comparison of survival status

Log-Rank test was used to compare survival time between categories of different predictors. Based on this test, survival time among different groups of predictors such as age, sex, insurance status, residence, nutritional status, and other baselines clinical and therapeutic characteristics were significantly different in survival time at a 5% level of significance (**Tables 1 & 3**).

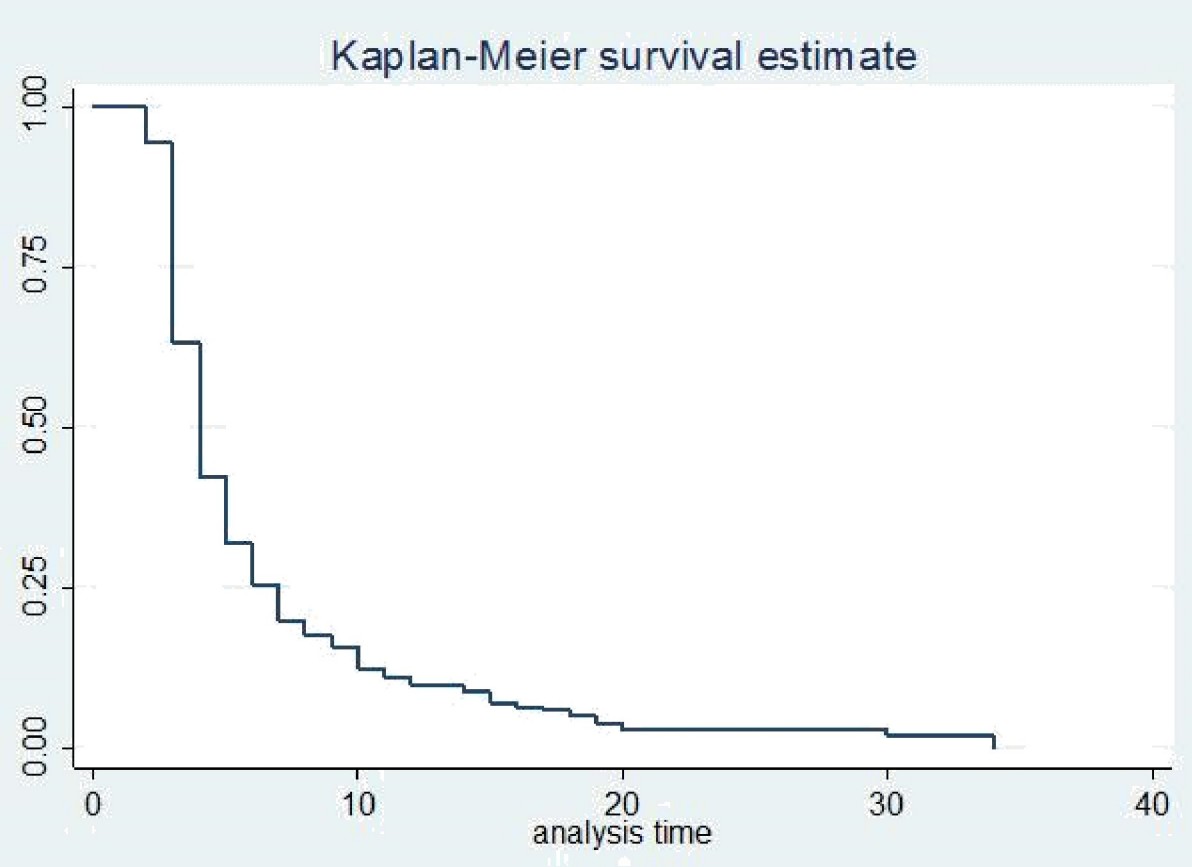

**Fig 1. Kaplan-Meier survival estimate of recovery time among children with SCAP admitted to Debre Markos referral hospital, 2016 to 2018.**

### Predictors of recovery time from SCAP

Predictors that had association at a p-value of ≤ 0.25 in bivariable Cox regression were included in multivariable Cox regression. Age, sex, stunting, vaccination status, duration prior to seeking care, danger sign associated with SCAP, co-morbidity, and drug regimen were selected through the stepwise backward elimination approach at a p-value of ≤ 0.25 level of significance. In the final stratified Cox regression model, seven predictors were analyzed and five predictors (age, stunting, duration of seeking care, and danger sign at admission and co-morbidity) became statistically significant predictors of recovery time from SCAP at a p-value of 0.05 level of significance (**Table 4**).

Age was an independent socio-demographic predictor. For a unit increase in age, the rate of recovery early from SCAP decreased by 6% (AHR; 0.94, 95% CI (0.90–0.98)). The nutritional status of children was seen in the analysis. The recovery rate of stunted children from SCAP decreased by 38% as compared to none stunted(AHR; 0.62, 95% CI (0.43–0.91)). The rate of recovery among children who sought care after five days of illness decreased by 36% as compared to their counterparts (AHR; 0.64, 95% CI (0.47–0.88)). The recovery rate of children who were admitted with danger sign reduced by 39% than those admitted without danger sign (AHR; 0.61, 95% CI (0.40–0.94)). Co-morbidity was an independent predictor of recovery time from SCAP among pediatrics. The rate of recovery among children admitted with co-morbidity decreased by 55% as compared to those children who had not co-morbidity at admission (AHR; 0.45, 95% CI (0.35–0.58)).

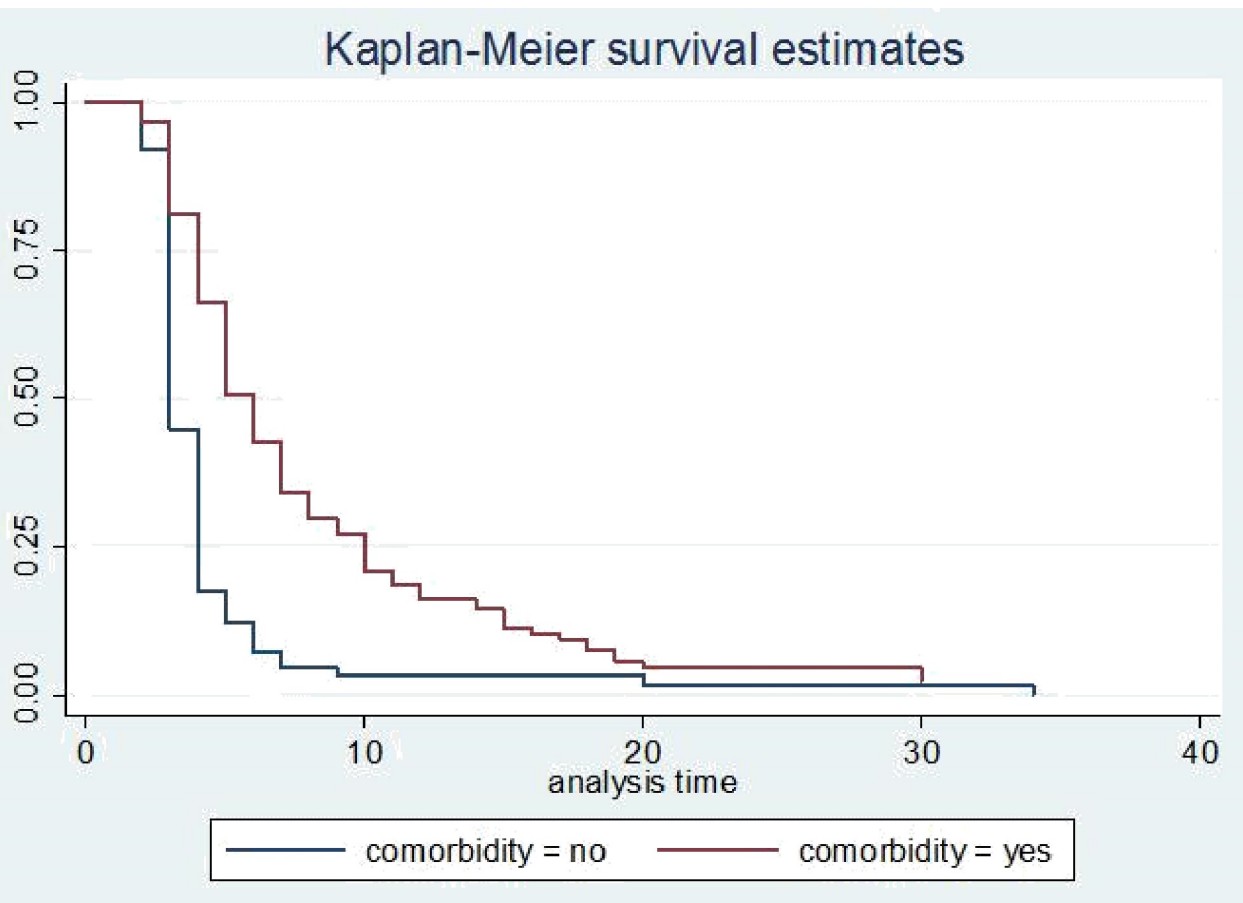

**Fig 2. Kaplan-Meier survival estimate for time to recovery among SCAP children with comorbidity.**

## Discussion

The overall median recovery time from SCAP was 4 days. In a study conducted in the southwest part of Ethiopia at Jimma, the duration of hospital stay was less than 3 for 76% of children which is almost consistent with what was observed in this study [15]. The finding of this study is almost similar to the study conducted in the rural health center of the Gambia which reported that the meantime of recovery was 4.5 days [16]. The finding of this study is higher than the study done at Vanderbilt (2.3days)and Nepal (2days) [9, 17]. This variation might be due to case-mix, severity, and co-morbidity differences.

This finding is lower than the study finding in Poland on trends in the hospitalization of children with bacterial pneumonia that reported 8.2 to 10.1 days [18]. This discrepancy might be due to the time difference as it was conducted from 2007 to 2011.

In this study, the age of children was an important socio-demographic predictor that has a significant effect on the recovery time of SCAP patients. This study indicated that younger children recover sooner than older ones. This finding is in line with the study conducted on time to clinical stability among hospitalized children that reported higher hospital stay among children above five years than below [9]. The current finding is also supported by a study conducted in Italy on the prediction of delayed recovery from pediatric community-acquired pneumonia [19]. This finding is not consistent with other study findings [17, 18]. A study on the trend of hospitalization of children revealed that the longest and the shortest hospital stay

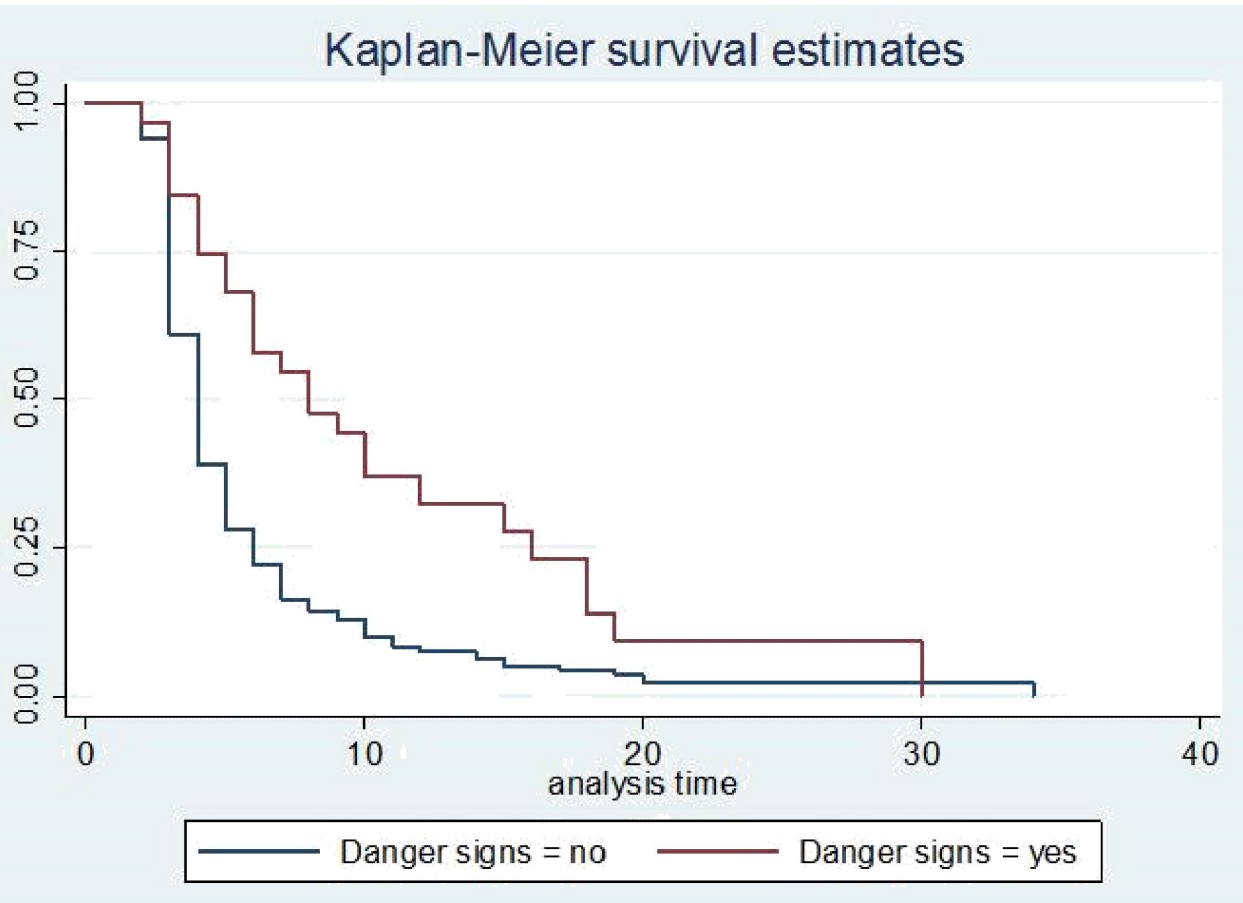

**Fig 3. Kaplan-Meier survival estimate for time to recovery among SCAP children with danger signs.**

was recorded among younger and older children respectively [18]. A similar study conducted in Nepal also showed that as age increases recovery time from illness also increases [17]. This discrepancy might be due to the difference in the age group of children as the studies included under-six and under-three years of children. Another possible reason for such contradiction could be as a result of variation in baseline clinical conditions and treatment protocols in the study areas.

The nutritional status of children who were admitted to the hospital due to SCAP has been investigated in the study. Stunting was significantly associated with the recovery time from the disease. Children who had stunting recovered slowly as compared to normal children.

This is due to the fact that malnutrition magnifies the effect of disease and cause more severe disease episodes, complications, and spends more time with illness [20]. Under-nutrition is often associated with the reduction of an immune system; consequently worsens the prognosis of the disease and prone to difficultly to recover [21].

The other predictor that has a significant effect in the median time to recovery from SCAP was danger sign at admission. The rate of recovery from SCAP was lower among patients who had danger signs at admission. This finding is in line with other studies [16, 17]. As children develop danger signs, the disease becomes severe, and the required time to recover from it becomes too long due to the pathophysiologic state of the disease [2].

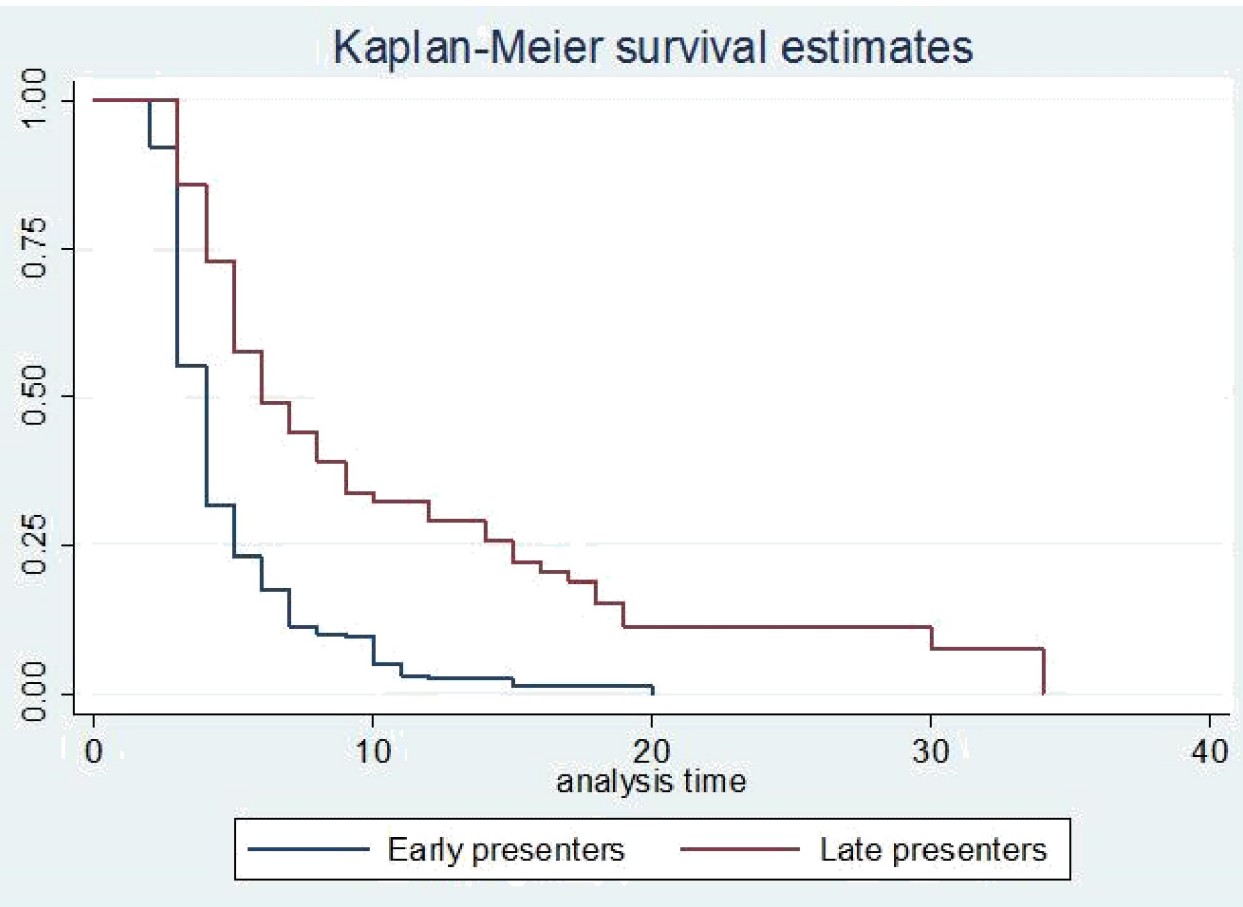

**Fig 4. Kaplan-Meier survival estimate for time to recovery among SCAP children with duration to seek care.**

Duration prior to seeking care was an independent significant predictor for the recovery time of pneumonia. Children who presented to the hospital early (before five days of illness) recovered sooner than those children presented lately. This finding is consistent with a prospective study conducted in the Gambia [16]. This is due to the fact that as a progression of disease increases, the required time to recover from it also increases. The other important predictor which was significantly associated with recovery time from SCAP was the presence of co-morbidity. Children who were admitted in the hospital with co-morbidity recover slowly as compared to children who were admitted without co-morbidity. This finding is in line with another study [16]. This is because when children acquire different illnesses at a time, their immune system decreases significantly, thus delay the recovery time.

The findings of this study should be interpreted with these limitations. As the study was conducted retrospectively, it failed to include all variables particularly parental socio-demographic, socio-economic, and environmental characteristics which could be potential predictors of the outcome variable. The outcome of recovery may be influenced by a range of factors, including bed availability, and may not have been consistent throughout the study period. Censored observations due to death might underestimate the recovery time as they were more severely ill. The effect size(hazard ratio) of the stratification variable for the stratified Cox regression model was not calculated. In addition, lack of comparator and potential confounders were considered as limitations of this study.

**Table 4. Stratified Cox regression model stratified by drug regimen of SCAP patients at DMRH, North West Ethiopia, 2016 to 2018.**

| Covariate | AHR(95%CI) | P-value |
|---|---|---|
| **Age (years)** | 0.94(0.90–0.98 | 0.016* |
| **Sex** | | |
| Male | 1 | |
| Female | 1.23(0.98–1.55) | 0.071 |
| **Stunting** | | |
| No | 1 | |
| Yes | 0.62(0.43–0.91) | 0.016* |
| **Vaccination status** | | |
| None vaccinated | 1 | |
| Partially vaccinated | 1.22(0.73–2.03) | 0.43 |
| Fully vaccinated | 1.52(0.93–2.5) | 0.09 |
| **Duration (days)** | | |
| ≤5(early presenters) | 1 | |
| >5(late presenters | 0.64(0.47–0.88) | 0.007* |
| **Danger sign** | | |
| No | 1 | |
| Yes | 0.61(0.40–0.94) | 0.025* |
| **Co-morbidity** | | |
| No | 1 | |
| Yes | 0.45(0.35–0.58) | < 0.001* |

* Statistically significant at 0.05.

## Conclusions

This study assessed the median time to recovery from SCAP and potential predictors. The median time to recovery from SCAP was 4 days which was long according to the British Thoracic Society (BTS) guidelines for the management of SCAP [22]. Age of children, stunting, danger sign at admission, duration prior to seeking care, and co-morbidity were statistically significant predictors of recovery time from the disease in the final stratified Cox regression model. However, sex, residence, vaccination status, and previous history of ARTI were not associated statistically with the recovery time.

Measures to reduce recovery time from the disease should be strengthened. Parents or caretakers shall take their children to the health facility immediately when they become ill. Health care providers should give due attention to children with the identified predictors while treating them. Further study using a prospective design by including other parental variables that were not included in this study is advised to fill its limitation.

## Supporting information

**S1 Dataset.**
(DTA)

## Acknowledgments

We would like to thank Debre Markos University and Debre Markos referral hospital administrators and staff for their cooperation.

## Author Contributions

**Conceptualization:** Belayneh Mengist.

**Data curation:** Belayneh Mengist.

**Methodology:** Belayneh Mengist.

**Software:** Belayneh Mengist.

**Writing – original draft:** Belayneh Mengist.

**Writing – review & editing:** Mulugeta Tesfa, Bekalu Kassie.

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
