## [Decision Letter · Decision Letter 0]

29 Apr 2020

PONE-D-20-03326

Time to recovery and its predictors among severe community acquired

pneumonia patients admitted to pediatric ward, Debre-Markos referral

hospital, North West Ethiopia, retrospective follow-up study.

PLOS ONE

Dear Mr Mengist,

Thank you for submitting your manuscript to PLOS ONE. After careful consideration, we feel that it has merit but does not fully meet PLOS ONE’s publication criteria as it currently stands. Therefore, we invite you to submit a revised version of the manuscript that addresses the points raised during the review process.

Two content expert peer reviewers have provided extensive comments around the opportunities to evolve the presentation of the work, both in terms of content and delivery. The analysis is something to be reviewed and strengthened as able, which in turn may impact the discussion.

We would appreciate receiving your revised manuscript by Jun 13 2020 11:59PM. To enhance the reproducibility of your results, we recommend that if applicable you deposit your laboratory protocols in protocols.io, where a protocol can be assigned its own identifier (DOI) such that it can be cited independently in the future. For instructions see: http://journals.plos.org/plosone/s/submission-guidelines#loc-laboratory-protocols

We look forward to receiving your revised manuscript.

Kind regards,

Shane Patman, PhD

Academic Editor

PLOS ONE

4. Please include your tables as part of your main manuscript and remove the individual files. Please note that supplementary tables (should remain/ be uploaded) as separate "supporting information" files

Reviewers' comments:

Reviewer's Responses to Questions

**Comments to the Author**

1. Is the manuscript technically sound, and do the data support the conclusions?

Reviewer #1: Partly

Reviewer #2: Partly

2. Has the statistical analysis been performed appropriately and rigorously? 

Reviewer #1: Yes

Reviewer #2: No

3. Have the authors made all data underlying the findings in their manuscript fully available?

Reviewer #1: No

Reviewer #2: No

4. Is the manuscript presented in an intelligible fashion and written in standard English?

Reviewer #1: No

Reviewer #2: Yes

5. Review Comments to the Author

Reviewer #1: This is a retrospective review of children admitted to an Ethiopian hospital with severe community acquired pneumonia. It has regional relevance and potential relevance to other countries with a similar socio-economic profile. I have several suggestions to further improve the article's content and external generalisability.

Major comments:

The article must include a better description of context and standard of care, as well as available resources. What are the average numbers of paediatric admissions per annum, and the overall mortality rate? Is PICU available, what level of respiratory/ventilatory support is available on the ward or in the hospital/region (e.g low/high flow oxygen therapy; non-invasive ventilation (CPAP/BiPAP); or invasive ventilation)? What proportion of children received any respiratory support? What antibiotic approach is standardly employed in terms of empiric treatment, and antibiotic stewardship – again, I presume the WHO guidelines were employed, but this should be stated. The only information in this regard is that just over half the children in the study received penicillin. What treatment did the remainder receive?

The above is highly relevant in the discussion section – studies cited include those from Europe, where there is likely to be high levels of intensive care provision, which might explain the longer duration of hospital stay (recovery time), which could be skewed by increased ICU admission and days receiving mechanical ventilation (balanced possibly by lower mortality). In the discussion there is vague reference to “differences in treatment and caring practices, health care settings and other socioeconomic factors between areas where the studies were conducted”. If the setting and healthcare practice at the study site is better described at the outset, then direct reference can be made to the possible reasons for discrepancies in findings.

Please provide the case definition of severe CAP - I presume the WHO definition was used, but this should be explicit please. Was any attempt made to differentiate CAP to other LRTI such as bronchiolitis?

Other comments:

Ethics review is indicated as “N/A” on the meta-data form, however in the text it is stated that institutional REC approval was obtained (as appropriate)

Throughout the article there are a number of grammatical errors, and in general the article is quite cumbersome, verbose and difficult to read. I suggest the article undergo language editing before resubmission. There is also a lot of unnecessary repetition of data in tables and text.

What were the organisms associated with CAP – bacterial and viral? Were cultures and viral samples sent for laboratory analysis? Aetiology is important to consider to ensure appropriate antibiotic cover.

Please list all the symptoms/signs classified as “Danger signs” – were these limited to reduced LOC, abnormal body movement and vomiting? What about hypoxia (SaO2 or presence of cyanosis), convulsions, inability to feed and presence of severe respiratory distress/failure?

Was oxygen saturation recorded? Hypoxia on admission has repeatedly been shown to be associated with worse outcome, and monitoring SaO2 is recommended in low resourced countries. I think these data should be presented separately and together, rather than only using the broad category “danger signs” – it is useful to know the relative proportions of children with SCAP presenting with signs of very severe CAP.

I would like to see acute and chronic comorbid conditions presented separately (can be combined if appropriate for regression analysis).

Throughout the Results section, please present full data , using n and percent rather than only using percentages. Similarly, where continuous data are presented in the text please present both the central tendency and the data spread (eg IQR).

Table 1 is superfluous, can delete – the data is presented again in Table 3 – just need to add the percentages under the column “Frequency”.

Figure 1 is not clear and probable not needed – rather present the n (%) outcome categories in the text. What outcome/s does the category “others” refer to?

P10 - Under the heading “Comparison of Survival Time”, please refer in the text to Tables 3 and 4.

Separate headings are given for “Survival Time” and “Time to Recovery”, however these are synonymous (according to case definitions provided). Please stick to one term, to avoid confusion.

P11 – it is not necessary to repeat data presented in the Tables, especially with the lack of precision currently given in the text. Either present all data (median, IQR and significance levels), or simply refer to the tables.

What was the median time to death in those who died?

The statistical analysis sections on pp11-12 I found quite long and difficult to read. I think this could be simplified. Please use full sentences throughout. Eg P12: “Likelihood ratio test which compares log-likelihood statistics for the interaction model and the no-interaction model”. What is this phrase referring to?

The discussion is very long with quite a bit of repetition, and could be shortened without losing important content. Where possible, please refrain from generalizing. On p 16, for example, stating that the finding that children presenting with comorbid conditions recover more slowly is because children with different illnesses take a prolonged time to recover is providing a circular argument for the observed result, it is not explaining the finding at all. Again, if acute and chronic comorbid conditions are reported separately, then more specific arguments can be made, for example related to immunosuppression (HIV infected children). The statement that comorbidity might lead to poor adherence to prescribed drugs leading to prolonged hospitalization is not supported by the study findings, and should be deleted. I also do not see how drugs prescribed and administered in a controlled hospital environment could be affected by “non-adherence”?

There are many more limitations inherent to a retrospective study, which have not been considered here. The one mentioned, that of incomplete medical records, as in fact an initial exclusion criterion, and therefore would not have impacted on the final study findings. The issues of lack of comparators and potential confounders impacting on outcome measures are important potential limitations. For example, when only 31 beds are available for all paediatric admissions, during seasonal increases in other infections (eg gastroenteritis), there may be different admission and discharge criteria, impacting on the “recovery” timepoint.

The conclusion for the first time mentions the BTS guidelines (without citation) – the long time to recovery must be discussed in this regard before the concluding statement.

Reviewer #2: Pneumonia continues to be a burden and clinical phenotypes may have changed following PCV introduction, hence this is an important topic.

The manuscript test is clear and well-written, but the authors should avoid giving text as bullet points. However, there are quite a few areas where more clarity and rigour over participants, definitions and statistical methods are needed.

Under ‘source population’ it is important to define the diagnostic criteria so that other centres may compare their results. If it is simply the clinician diagnosis, it is important to say if this was determined at the admission or at discharge. However using only the clinician diagnosis would significantly weakening the article.

Under the sample size section please give the prevalence and effect size that was being targeted in order to justify the sample size calculation.

Since time to discharge was used as the primary endpoint, ’length of stay’ or something similar should be used in the title and throughout the text rather than referring to ‘recovery’ since other factors may have altered the hospitalisation duration, especially when severe malnutrition or co-morbidities were present.

Please add to whether or not data on the specific features after severe pneumonia as defined by the World Health Organisation (and the BTS) were available, and why they are not presented in the independent variables. This is important because the signs are used to define with the severity of pneumonia.

Among the independent variables please indicate whether continuous values for the anthropometry were used and how they were converted into Z scores.

For the Cox regression models two methods of variable selection are given bivariate testing of individual variables and a backward stepwise method. Please explain whether these were done in order or whether both were done on the raw data. Please provide the probability for removal criteria used for the backward stepwise selection. Please provide the rationale and the definitions of the strata.

In the methods please describe how the status of fully vaccinated or partially vaccinated was defined. One problem with this variable is that it is age-dependent and since only children of nine months or more would be eligible for measles vaccine for example.

Please also defined terms used in results including hyperactive airways disease and high-grade fever.

One problem with examining both time to discharge and survival time is that they are competing risks. For example, if a child dies they would not be eligible to have their time to discharge recorded. The usual way to deal with this problem is to undertake competing risks analysis. An example is children with a danger sign, who have a higher risk of mortality for whom the effect on time to discharge could be biased.

With only 39 events, reporting the risk factors for mortality is likely to be misleading if many independent variables were tested. It would be better to focus on time to discharge.

Season effects may be important. RSV is typically the leading cause of pneumonia, it is usually highly seasonal.

In the tables, some of the 95%Cis are missing. However, it is likely these should be interquartile ranges as time to discharge is skewed (this is correct in the manuscript text).

I wasn’t clear what is being tested in the ‘checking no interaction assumption’. If all possible interactions are being tested then this needs to be described in the methods section. The assumption in a Cox model is that effects are not time-varying

One point on terminology when reporting the Cox model, the aHR for a feature gives the how much more or less likely a child is to be discharged every day rather than the ‘rate of recovery’.

6. PLOS authors have the option to publish the peer review history of their article (what does this mean?). If published, this will include your full peer review and any attached files.

Reviewer #1: No

Reviewer #2: No

---

## [Author Response · Author response to Decision Letter 0]

20 May 2020

Per the given comments and recommendations, a separate file which addresses comments and suggestions of the reviewers and academic editor is submitted as 'response to reviewers'

---

## [Decision Letter · Decision Letter 1]

19 Jun 2020

PONE-D-20-03326R1

Time to recovery and predictors of severe community acquired pneumonia among pediatric patients in Debre Markos referral hospital, North West Ethiopia: a retrospective follow-up study

PLOS ONE

Dear Dr. Mengist,

Thank you for submitting your manuscript to PLOS ONE. After careful consideration, we feel that it has merit but does not fully meet PLOS ONE’s publication criteria as it currently stands. Therefore, we invite you to submit a revised version of the manuscript that addresses the points raised during the review process.

At these uncertain times around the world I was fortuitous in engaging the same two peer reviewers that contributed to the initial manuscript review to contribute to the review of this revised submission. Whilst they note some positive evolving with this resubmission, it is felt that further work is still required to enhance the reporting of your word. Further constructive comments are included below. Importantly it is felt that the input of a scientific writer with English as a first language would add value to improving the readability of your work; syntax and grammar are both areas that need attention.

We look forward to receiving your revised manuscript.

Kind regards,

Shane Patman, PhD

Academic Editor

PLOS ONE

Reviewers' comments:

Reviewer's Responses to Questions

**Comments to the Author**

1. If the authors have adequately addressed your comments raised in a previous round of review and you feel that this manuscript is now acceptable for publication, you may indicate that here to bypass the “Comments to the Author” section, enter your conflict of interest statement in the “Confidential to Editor” section, and submit your "Accept" recommendation.

Reviewer #1: (No Response)

Reviewer #2: (No Response)

2. Is the manuscript technically sound, and do the data support the conclusions?

Reviewer #1: Partly

Reviewer #2: No

3. Has the statistical analysis been performed appropriately and rigorously? 

Reviewer #1: Yes

Reviewer #2: No

4. Have the authors made all data underlying the findings in their manuscript fully available?

Reviewer #1: Yes

Reviewer #2: Yes

5. Is the manuscript presented in an intelligible fashion and written in standard English?

Reviewer #1: No

Reviewer #2: No

6. Review Comments to the Author

Reviewer #1: Thank you for inviting me to review the revision of this manuscript. In general, I find it to be much easier to read and the authors have taken most review comments under consideration. I have one major and minor comments related to this work.

Major comments

I am concerned about the outcome of “recovery”, which is highly subjective, being determined simply from clinician assessment. This is likely influences by a range of factors, including bed availability vs. demand, and may not have been consistent throughout the study period. This should be discussed and included as a limitation.

Minor comments:

Throughout the manuscript there is still clear evidence that the manuscript has not been edited by a native English speaker. Examples are given below, but after page 7 I stopped looking at grammar, as I believe this is beyond the scope of scientific review.

P3: “The hospital received 3293 children per year in average with the available 31 beds” change to “:The hospital admits an average of 3293 children per year, with 31 available beds”.

P3: “children with severe community acquired pneumonia were admitted in pediatric ward and further diagnose and treatment were provided by pediatricians, general practitioners and nurses” – change to “children with severe community acquired pneumonia

were admitted into the pediatric ward and further diagnosis and treatment were provided by

pediatricians, general practitioners and nurses”.

P5: “were admitted by sever community acquired pneumonia” – should be “admitted with severe CAP”

P5 – the medical charts were not extracted, presumably data were extracted from the medical charts.

P6 - “…inability to feed accompanying with SCAP” – change to “…inability to feed in addition to SCAP”

P7 – “fully vaccination was defined…” should be “full vaccination status was defined”

P7 – Very clumsy paragraph: “The patient records were first observed and an appropriate data extraction format was prepared in English. Data collectors used the data collection tool to collect the information from children’s charts. Charts were retrieved using the children’s registration number which was found in the database in the electronic system and the data clerks of the hospital were also support by identifying the charts”.

This can be rewritten much more clearly and simply, for example: “Patient records were retrieved using their unique registration number identified in the electronic hospital database. Charts and records were reviewed and data were extracted using a standardised English data extraction form”.

Reviewer #2: Some changes have been made but inclusion criteria are still not clear. The signs for SCAP and danger signs should be given in tables of characteristics, even if grouped for analysis. The methods of analysis are still unclear.

In methods:

It would be helpful to include the definition of SCAP in the eligibility criteria rather than later under operational definitions. I am concerned that these essential features as well as individual danger signs and comorbidities do not appear in the baseline characteristics tables or analysis since they are likely to be major predictors of time to recovery.

I did not understand this section: “Predictor drug regimen violates the proportional hazard assumption and the Cox-Snell residual suggests that the Cox proportional hazard model (Cox regression) does not fit the data adequately. Therefore an extension of the Cox proportional hazard model is required for this data. Since predictors were tested as not time-dependent, the stratified Cox regression model was adequate to be used”. Was drug treatment time-dependent or not? If the model stratified by drug treatment then how is the hazards ratio for drug treatment calculated (I note the log rank test was highly significant)? The usual way to deal with this in common software packages is to declare the variable as a time-varying covariate (e.g. using the tvc() option with stcox in STATA, this does not require repeated measures).

In the next paragraph, what interaction was tested? Was it an interaction with time to deal with the violation of the proportional hazard assumption?

In results:

I agree with not examining risks for mortality. There will be some bias due to deaths and children leaving against advice if the 11% censored were more severely unwell for example. I should at least be mentioned as a limitation.

Should the heading “Incidence rate of SCAP” be “Incidence of recovery from SCAP”?

The terms “survival time of SCAP”, “survival time” and “The survival status of children with SCAP was estimated by...” are confusing (although possibly statistically correct) since recovery rather than survival is being studied. “Time to recovery” would be better.

In the discussion, case mix, severity and comorbidity differences are likely to underlie differences to other studies. Hence also the need to describe these for this study.

There is still a need to run a spelling and grammar check (e.g. “Anthrophometric”, “vaccins”, “ceftriaxon”, “”). There are also random capital letters: “In this study, Hyperactive airway disease...”, “...study conducted in Italy on the Prediction of delayed recovery from...” etc.

7. PLOS authors have the option to publish the peer review history of their article (what does this mean?). If published, this will include your full peer review and any attached files.

Reviewer #1: No

Reviewer #2: No

---

## [Author Response · Author response to Decision Letter 1]

24 Jul 2020

We would like to thank you for your constructive comments. We have addressed comments and suggestions of the reviewers and academic editor.

---

## [Decision Letter · Decision Letter 2]

11 Aug 2020

PONE-D-20-03326R2

Time to recovery and predictors of severe community-acquired pneumonia among pediatric patients in Debre Markos referral hospital, North West Ethiopia: a retrospective follow-up study

PLOS ONE

Dear Dr. Mengist,

Thank you for submitting your manuscript to PLOS ONE. After careful consideration, we feel that it has merit but does not fully meet PLOS ONE’s publication criteria as it currently stands. Therefore, we invite you to submit a revised version of the manuscript that addresses the points raised during the review process.

We look forward to receiving your revised manuscript.

Kind regards,

Shane Patman, PhD

Academic Editor

PLOS ONE

Reviewers' comments:

Reviewer's Responses to Questions

**Comments to the Author**

1. If the authors have adequately addressed your comments raised in a previous round of review and you feel that this manuscript is now acceptable for publication, you may indicate that here to bypass the “Comments to the Author” section, enter your conflict of interest statement in the “Confidential to Editor” section, and submit your "Accept" recommendation.

Reviewer #2: (No Response)

2. Is the manuscript technically sound, and do the data support the conclusions?

Reviewer #2: Partly

3. Has the statistical analysis been performed appropriately and rigorously? 

Reviewer #2: No

4. Have the authors made all data underlying the findings in their manuscript fully available?

Reviewer #2: Yes

5. Is the manuscript presented in an intelligible fashion and written in standard English?

Reviewer #2: No

6. Review Comments to the Author

Reviewer #2: There are improvements but the case definition characteristics ‘oxygen saturation <90% or central cyanosis or severe respiratory distress or inability to drink or breastfeed or vomiting everything, altered consciousness, and convulsions’ still need to be tabulated. Every child must have had at least one of these because they formed the case definition, so it is not clear why they are not analysed.

In the discussion, case mix rather than ‘socio-economic differences in the study areas’ seems a more likely explanation for differences between studies.

There are areas that still need language editing, for example:

P9: ‘308 (87.5%) fed breast for the first six months exclusively.’

P10: ‘313 (88.9%) children develop the event’

P17: ‘Stunted children recovered from the disease slower in compression to normal

children.’

7. PLOS authors have the option to publish the peer review history of their article (what does this mean?). If published, this will include your full peer review and any attached files.

Reviewer #2: No

---

## [Author Response · Author response to Decision Letter 2]

18 Aug 2020

We would like to thank you for your constructive comments and timely response. We addressed comments and suggestions of the reviewer and academic editor in the revised manuscript.

---

## [Decision Letter · Decision Letter 3]

2 Sep 2020

PONE-D-20-03326R3

Time to recovery and predictors of severe community-acquired pneumonia among pediatric patients in Debre Markos referral hospital, North West Ethiopia: a retrospective follow-up study

PLOS ONE

Dear Dr. Mengist,

Thank you for submitting your manuscript to PLOS ONE. After careful consideration, we feel that it has merit but does not fully meet PLOS ONE’s publication criteria as it currently stands. Therefore, we invite you to submit a revised version of the manuscript that addresses the points raised during the review process.

We look forward to receiving your revised manuscript.

Kind regards,

Shane Patman, PhD

Academic Editor

PLOS ONE

Additional Editor Comments (if provided):

Significant positive progress is evident and the manuscript is on the verge of acceptance. Reviewer 2 has offered a few pertinent recommended revisions for consideration, which hopefully will not take too much time or effort to consider for incorporation.

Reviewers' comments:

Reviewer's Responses to Questions

**Comments to the Author**

1. If the authors have adequately addressed your comments raised in a previous round of review and you feel that this manuscript is now acceptable for publication, you may indicate that here to bypass the “Comments to the Author” section, enter your conflict of interest statement in the “Confidential to Editor” section, and submit your "Accept" recommendation.

Reviewer #1: All comments have been addressed

Reviewer #2: (No Response)

2. Is the manuscript technically sound, and do the data support the conclusions?

Reviewer #1: Yes

Reviewer #2: Yes

3. Has the statistical analysis been performed appropriately and rigorously? 

Reviewer #1: Yes

Reviewer #2: Yes

4. Have the authors made all data underlying the findings in their manuscript fully available?

Reviewer #1: Yes

Reviewer #2: Yes

5. Is the manuscript presented in an intelligible fashion and written in standard English?

Reviewer #1: Yes

Reviewer #2: Yes

6. Review Comments to the Author

Reviewer #1: I have no further comments, authors responded satisfactorily to a previous review.

Reviewer #2: The authors have made extensive revisions which have improved the manuscript. Defining SCAP and treatment failure is very helpful. The GLM model is an improvement and the signs associated with failure at 48h (convulsions, cyanosis, hypoxia, HIV, MAM & SAM) appear to be appropriate and in line with prior studies.

There are a few areas that could be further improved:

Major comment: in figure 1, please indicate deaths before 48h separately. This is also needed in Results at around line 202 since it is the most important outcome.

Please mention in the methods that children who immediately started second-line antibiotics were excluded (as per the response to reviewers) and give the number in results.

In figure 1, exclusions can be simply described as “Not meeting SCAP criteria or out of the study age range”.

At line 178, please move “Respiratory distress in this study context was considered as the presence of grunting and head nodding” and “In this study, an acute form of malnutrition was assessed among the participants using the WHO Z-score charts of weight for height or length. Z-scores of between -3SD and ≤ -2SD were equated to moderate acute malnutrition and Z-scores of ≤ -3SD were equated to severe acute malnutrition, also the presence of kwashiorkor was considered as severe acute malnutrition.” to the Methods section.

It is important to note that previous studies may have used the WHO 2005 pocketbook definition of severe pneumonia which is much less severe than the 2013 version (2013 version is equivalent to the 2005 ‘very severe pneumonia’ syndrome). This applies to “studies done in Kenya (10) and South Africa (14)” at line 238. In the Kenyan study, ‘very severe pneumonia’ had a failure rate of around 30%. I am not sure if any of the South African children would have met the 2013 severe criteria since only amoxicillin treatment was given.

Please comment on the frequency of x-ray findings in comparison with other studies such as PERCH.

Please check spellings and spaced throughout e.g. “Amipicilin” at line 237, “...gram-negative bacteria are resistant to ampicillin and gentamycin...” at line 272 should be “...Gram-negative bacteria are resistant to ampicillin and gentamicin...”. Space is missing at lines 114 “6hourly” and 210-211 “p value<0.001”, please check for others.

7. PLOS authors have the option to publish the peer review history of their article (what does this mean?). If published, this will include your full peer review and any attached files.

Reviewer #1: No

Reviewer #2: **Yes: **James A Berkley

---

## [Author Response · Author response to Decision Letter 3]

7 Sep 2020

We would like to thank you for your constructive comments and timely response. We have addressed comments and suggestions of the reviewer and academic editor.

---

## [Editor Report · Decision Letter 4]

11 Sep 2020

Time to recovery and predictors of severe community-acquired pneumonia among pediatric patients in Debre Markos referral hospital, North West Ethiopia: a retrospective follow-up study

PONE-D-20-03326R4

Dear Dr. Mengist,

We’re pleased to inform you that your manuscript has been judged scientifically suitable for publication and will be formally accepted for publication once it meets all outstanding technical requirements.

Kind regards,

Shane Patman, PhD

Academic Editor

PLOS ONE
---

## [Editor Report · Acceptance letter]

15 Sep 2020

PONE-D-20-03326R4 

Time to recovery and predictors of severe community-acquired pneumonia among pediatric patients in Debre Markos referral hospital, North West Ethiopia: a retrospective follow-up study 

Dear Dr. Mengist:

I'm pleased to inform you that your manuscript has been deemed suitable for publication in PLOS ONE. Congratulations! Your manuscript is now with our production department. 

Kind regards, 

on behalf of

Assoc Prof Shane Patman 

Academic Editor

PLOS ONE